# Update on Glycosphingolipids Abundance in Hepatocellular Carcinoma

**DOI:** 10.3390/ijms23094477

**Published:** 2022-04-19

**Authors:** Frances L. Byrne, Ellen M. Olzomer, Nina Lolies, Kyle L. Hoehn, Marthe-Susanna Wegner

**Affiliations:** 1School of Biotechnology and Biomolecular Sciences, University of New South Wales, Sydney, NSW 2052, Australia; frances.byrne@unsw.edu.au (F.L.B.); e.olzomer@unsw.edu.au (E.M.O.); k.hoehn@unsw.edu.au (K.L.H.); 2Pharmazentrum Frankfurt/ZAFES, Institute of Clinical Pharmacology, Johann Wolfgang Goethe University, Theodor Stern-Kai 7, 60590 Frankfurt, Germany; ninaschoemel@gmx.de

**Keywords:** glycolysis, GEMs, oxidative phosphorylation, UGCG, glucosylceramide, normal liver cells, globosides, gangliosides, lacto/neo-lacto series GSLs, sulfatides

## Abstract

Hepatocellular carcinoma (HCC) is the most frequent type of primary liver cancer. Low numbers of HCC patients being suitable for liver resection or transplantation and multidrug resistance development during pharmacotherapy leads to high death rates for HCC patients. Understanding the molecular mechanisms of HCC etiology may contribute to the development of novel therapeutic strategies for prevention and treatment of HCC. UDP-glucose ceramide glycosyltransferase (UGCG), a key enzyme in glycosphingolipid metabolism, generates glucosylceramide (GlcCer), which is the precursor for all glycosphingolipids (GSLs). Since UGCG gene expression is altered in 0.8% of HCC tumors, GSLs may play a role in cellular processes in liver cancer cells. Here, we discuss the current literature about GSLs and their abundance in normal liver cells, Gaucher disease and HCC. Furthermore, we review the involvement of UGCG/GlcCer in multidrug resistance development, globosides as a potential prognostic marker for HCC, gangliosides as a potential liver cancer stem cell marker, and the role of sulfatides in tumor metastasis. Only a limited number of molecular mechanisms executed by GSLs in HCC are known, which we summarize here briefly. Overall, the role GSLs play in HCC progression and their ability to serve as biomarkers or prognostic indicators for HCC, requires further investigation.

## 1. Background

Liver cancer is the third leading cause of cancer-related deaths worldwide [1]. In 2020, there were more than 900,000 new cases and 800,000 deaths attributed to this malignancy [1]. Unsettlingly, the incidence of this cancer type is growing, and it is estimated that more than 1 million new cases will be diagnosed per year worldwide by 2025 [2]. Hepatocellular carcinoma (HCC) is the most frequent type of primary liver cancer, accounting for 75–90% of all cases [1,2]. The greatest risk factors for HCC are viral infections by hepatitis B, which accounts for more than half of all liver cancer cases and deaths, and hepatitis C [1,2]. In the West, hepatic injury leading to non-alcoholic steatohepatitis (NASH) is becoming a more prominent risk factor for HCC due to increased rates of obesity, diabetes, and metabolic syndrome [2]. Treatment options for advanced HCC (which accounts for 40% of HCCs at diagnosis) are limited to supportive care and systemic therapies including the multi-tyrosine kinase inhibitors sorafenib, lenvatinib, cabozantinib, and regorafenib [3,4]; immune checkpoint inhibitors; and monoclonal antibodies [2]. The recently approved combination therapy employing antibodies against programmed cell death 1 ligand 1 (atezolizumab) and vascular endothelial growth factor (bevacizumab) has led to a marked improvement in progression-free and overall survival compared with the standard of care agent, sorafenib [2]. Despite increased incidence of serious adverse events (reviewed in [5]), atezolizumab plus bevacizumab is currently the first-line treatment for HCC patients (reviewed in [6]). Although novel therapeutic approaches have been developed, palliative care applies to many HCC patients. Therefore, investigating the molecular mechanisms underpinning HCC initiation and progression may provide vital clues to aid the development of novel therapeutic strategies for the prevention and treatment of HCC. As such, new therapeutics for the treatment of liver cancer are highly desired and are a global health priority.

Glycosphingolipids (GSLs) contain lipid and sugar moieties. They are important components of the cellular membrane and act as signaling molecules in cellular processes such as apoptosis [7]. *Uridine diphosphate (UDP)-glucose ceramide glucosyltransferase* (UGCG) catalyzes the first glycosylation step in the synthesis of GSLs. By transferring UDP-glucose to ceramide, glucosylceramide (GlcCer, cerebroside) is produced, which is the precursor for all complex GSLs. Therefore, UGCG, which resides in the Golgi apparatus, is the key enzyme of GSL metabolism. For more detailed insight into GSL production, we refer to our review from 2020 [8]. By adding a galactose molecule to GlcCer, lactosylceramide (globoside) is synthesized. More complex GSLs, known as globosides, such as globotriaosylceramide (Gb3) and gangliosides such as monosialodihexosylganglioside (GM3) are produced by adding monosaccharides to lactosylceramide (reviewed in [9]). The precursor for sulfatide (glycosphingolipid sulfate) is galactosylceramide, which is produced in the endoplasmic reticulum (ER). Sulfatides carry a sulfate ester group attached to the carbohydrate moiety.

Previous studies have demonstrated that UGCG gene expression is altered in 0.8% of HCC tumors (TCGA, Firehose Legacy) [10,11], and UGCG overexpression is linked to multidrug resistance development in cancer cells (reviewed in [12]). In hepatoma cells, UGCG silencing or pharmacological inhibition restored cell sensitivity to sorafenib [13]. Accordingly, glycosphingolipids may play a role in HCC and resistance to sorafenib. With that said, it is surprising that only a limited number of studies have investigated the role of UGCG in HCC. Furthermore, most studies investigate complex GSLs, and only recently (since 2010) has there been an increased number of published studies relating to GSLs in HCC (Figure 1A). Of interest, novel therapeutic approaches for HCC patients such as artificial GSLs are currently under investigation. Okuda et al. showed that immunization with artificial GSLs leads to production of antibodies against α-fetoprotein-L3, which is an HCC-specific antigen [14]. Another study showed that combination therapy with an anti-PD-1 antibody and α-galactosylceramide leads to activation of dendritic cells in Hepa1-6-1 tumors resulting in checkpoint blockade [15]. Clinical trials are currently underway with α-galactosylceramide in other cancer types such as melanoma and lung cancer (reviewed in [16]).

In this review, we discuss UGCG in normal liver cells and mention briefly how UGCG and GSLs influence cells affected by Gaucher disease, an inherited genetic disorder caused by the deficiency of the enzyme *glucocerebrosidase* (GBA). Furthermore, we summarize how GSL species levels are altered in HCC. We will not cover chronic liver diseases such as NASH, which is an important driver of HCC, since this is already covered in depth here [19].

## 2. UGCG in Normal Liver Cells

To investigate early onset of UGCG-mediated pro-cancerous changes in normal liver cells, we overexpressed UGCG in NMuLi (normal murine liver) cells (NMuLi/UGCG OE) [18]. Overexpression (OE) in NMuLi cells leads to decreased mitochondrial respiration and was rescued by treatment with the UGCG inhibitor EtDO-P4 [18]. The effect was mediated through accumulation of UGCG-derived GlcCer and lactosylceramide in ER/mitochondria fractions, which induced mitochondrial superoxide [18]. Other studies support the finding that GSLs are related to mitochondrial dysfunction (reviewed in [8]). However, this effect might also be induced by dihydroceramide since dihydroceramide levels were elevated in ER/mitochondria fractions of UGCG OE liver cells. Siddique et al. showed that increased dihydroceramide impairs ATP production [20]. However, glycolysis was also decreased in NMuLi/UGCG OE cells (rescued following EtDO-P4 treatment), which might be mediated through decreased phosphorylation and subsequent deactivation of *AMP-activated protein kinase* (AMPK) α (P-Thr172). This leads to inhibition of energy producing pathways such as glycolysis [21]. In neural cells (primary astrocytes) activation of AMPK prevents de novo synthesis of ceramide which prevents *rapidly accelerated fibrosarcoma 1* (Raf-1)/extracellular signal-regulated kinase activation, and apoptosis induction [22]. Since AMPKα is less phosphorylated, we expected an increase in ceramide levels in NMuLi/UGCG OE cells. Interestingly, ceramide is lower in *glycosphingolipid-enriched microdomains* (GEMs) (fractions 2 and 3, GEMs verification published in [18]) of UGCG overexpressing normal liver cells compared to control cells (Figure 1B); disproving the described effect of AMPK on the de novo ceramide synthesis in, at least, normal liver cells. The reduction in ceramide levels might be due to increased GlcCer production leading to ceramide clearance. In addition, the expression levels of the *liver cancer stem cell* (LCSC) markers (reviewed in [23]) EPCAM, CD13, CD133, CD90.1, and CD44 were lower in NMuLi/UGCG OE cells compared to control cells [18]. Interestingly, mice without hepatic expression of acetyl-CoA carboxylase (executes the first step of de novo lipogenesis in the cytosol) develop liver tumors [24]. A connection between UGCG and acetyl-CoA carboxylase in murine fibroblasts has been shown by Ishibashi and Hirabayashi [25]. AMPK-induced phosphorylation and subsequent inhibition of acetyl-CoA carboxylase leads to decreased UGCG activity and lowered GlcCer levels. Inhibition of acetyl-CoA carboxylase lowers the production of malonyl-CoA, which is essential for fatty acid synthesis [26]. However, we could not detect a clear trend to determine whether UGCG and acetyl-CoA carboxylase are connected in the context of liver tumor development [18]. In summary, overexpression of UGCG/GlcCer in normal liver cells did not induce pro-cancerous cellular processes.

## 3. UGCG and Gaucher Disease

Mutation of the *glucocerebrosidase* (GBA) gene results in GlcCer accumulation and is described as the lysosomal storage disease *Morbus Gaucher* [27]. More than 450 gene mutations for *GBA1* have been identified. Gaucher disease patients exhibit heterogeneous phenotypes. The age of onset and the absence/presence or extent of neurological complications defines the clinical Gaucher disease type. Type 1 is defined as *non-neurological*, Type 2 as *acute neuronopathic,* and type 3 as *chronic neuronopathic* (reviewed in [28]). Morbus Gaucher patients exhibit hypermetabolism [27] (reviewed in [29]) and an increased risk for liver cancer [30]. Therefore, it is noteworthy to identify the role of GlcCer in liver cell metabolism and how GlcCer contributes to the pathology of liver tumors. Tumor development in Gaucher disease patients is linked to chronic cell and tissue inflammation. Accordingly, the immune system is dysregulated (reviewed in [31,32]). In addition, autophagic processes are changed in Gaucher disease patient cells (reviewed in [31]). Future studies with immune cell infiltration experiments following grafting of UGCG overexpressing cells in mice would be helpful to elucidate molecular mechanisms induced by UGCG. We recommend reading the review from Wątek et al. for detailed information about how GlcCer accumulation influences immunomodulatory functions and therefore contributes to carcinogenesis induction in Gaucher disease patient cells [33]. An alternative metabolic pathway leads to lyso-GlcCer (glucosylsphingosine) in a Gaucher disease mouse model (reviewed in [34]). We suggest reading the review from Stirnemann et al. for detailed information about how the biomarker lyso-GlcCer for Gaucher disease patients contributes to Gaucher disease pathology (reviewed in [34]).

## 4. Glycosphingolipids in HCC

### 4.1. UGCG/GlcCer

Previous studies have shown that UGCG expression is altered in 0.8% of HCC tumors (TCGA, Firehose Legacy) [10,11]. UGCG gene expression is increased in HCC tissue compared to non-cancerous tissue [10,11,34] and Li et al. showed upregulation of GlcCer (hexosylceramide) in serum samples of HCC patients [35] (Table 1). Interestingly, in a review by Simon et al. it was reported that during HCC development, cellular ceramide levels decrease, and *sphingosine-1-phosphate* (S1P) levels increase [19]. This decrease in ceramide supports tumor growth and inhibits apoptosis, which correlates with the high proliferative capacity of HCC (reviewed in [19]). Besides S1P production, ceramide clearance is also achieved by UGCG, which again poses the question of how GSLs are involved in HCC development. Jennemann et al. showed delayed tumor growth in *diethylnitrosamine* (DEN)-induced liver tumors in mice, which exhibit a liver specific UGCG *knockout* (KO) [36]. The effect is mediated by decelerated cytokinesis, but the precise molecular mechanisms are unknown. Interestingly, the sphingomyelin concentration is increased in normal liver and tumor tissues of liver specific UGCG KO mice, which could be a cellular mechanism to avoid ceramide induced apoptosis [36]. Since a lack of GSLs per se does not prevent liver tumor development, other proteins/pathways, beside UGCG, may play a role in HCC development. Indeed, UGCG mRNA expression and GlcCer levels are increased in the livers of mice with *mechanistic target of rapamycin* (mTOR)-activated HCC tumors, and UGCG inhibition reduced proliferation of hepatocytes, tumor burden, and markers of liver damage [37]. These data suggest that UGCG is indeed involved in tumor development, but only when carcinogenesis was already activated.

The role of UGCG in liver tumorigenesis is likely mTOR-dependent and mTORC2 might be a potential target to treat HCC. Another study showed sorafenib induced UGCG expression leading to sorafenib resistance in liver cancer cells [13]. However, following UGCG inhibition, no alterations in *phosphatidylinositol 3-kinase* (PI3K)/*protein kinase B* (AKT) and RAF)/*mitogen-activated protein kinase* (MAPK)/extracellular signal-regulated kinase signaling were detected in liver cancer cells [30]. These, compared to normal liver cells, are contradictory study results. This contradiction could be related to the induction of the mentioned signaling pathways rather in the onset of carcinogenesis than to the induction of these signaling pathways in established cancer cells. Interestingly, downregulation of *ORMDL sphingolipid biosynthesis regulator 3* (ORMDL3) in the human liver cancer cell line HepG2 led to increased levels of GlcCer (C16, C22, C24:0) and downregulation of ORMDL1 also led to an increase in GlcCer levels (C16, C20, C22, C24) to an even greater extent [58]. Furthermore, irritant-induced inflammation decreased ORMDL protein expression and increased GlcCer levels (C22, C24) in the livers of mice. These data indicate that ORMDLs may be involved in regulation of ceramides during interleukin-1-mediated sterile inflammation in liver cancer cells [58]. Ying et al. utilized data from the UALCAN web resource to show that the expression of *glucosylceramidase beta 3* (GBA3) is significantly decreased in HCC tissues [59], leading to GlcCer accumulation. In contrast to GBA1 (lysosomal) and GBA2 (extra-lysosomal), GBA3 is localized in the cytosol and exhibits its highest activity at neutral pH (GBA3 identified by Hayashi et al. [60]). GBA3 mRNA expression is significantly lower in HCC than in non-tumor liver tissue (328 HCC samples, 151 non-tumorous liver tissues) [59]. Furthermore, HCC patients with low GBA3 expression have a shorter survival time and a poor prognosis. Accordingly, high GBA3 expression (lower GlcCer levels) in HCC may favor a better prognosis.

### 4.2. Globoside Lactosylceramide

There is a limited number of published studies about lactosylceramide and how they impact HCC (Table 1). One study identified lactosylceramide as a biomarker in five types of HCC cell lines [38]. Another study analyzed serum samples of HCC patients and showed that lactosylceramide is increased [35]. Souady et al. also showed enrichment of lactosylceramide in cancerous liver tissue compared to healthy tissue [39]. Further research is needed to clarify whether this GSL species might be a novel therapeutic target in HCC.

### 4.3. Globosides (DSSG, Gb3, Gb2, Globo H, Gb4, iso-Gb4)

*Disialosyl galactosyl globoside* (DSGG), Gb3, and Gb2 are increased in the serum of HCC patients compared to the serum of healthy individuals [40]; therefore, they may serve as prognostic markers for this disease (Table 1). This is the first time DSGG has been shown to be highly expressed in HCC [40]. DSSG has been linked to metastatic potential in renal cell carcinoma [61]. Contradictory data from Souady et al. showed reduced expression of Gb3 and Gb4 in malignant liver tissue [39]. However, Globo H (Fucosyl-Gb5) might be a potential prognostic marker for HCC patients as well, since Zhu et al. showed that Globo H is only expressed in human HCC tissues, but not in peritumoral tissues [41]. Furthermore, Su et al. showed that Globo H is expressed on cancer stem cells generated from Hepa-1 cells [42], while Ariga et al. showed accumulation of iso-Gb4 in female rat hepatomas [43]. Thus, globosides may play a role in HCC development and serve as markers for malignant liver tissue.

### 4.4. Gangliosides (GM1, GM2, GM3, GD3, NeuGcGM3 Ganglioside)

Very early studies showed that GM2 levels are increased in two HCC samples compared to normal liver tissue [44,45] and that GM3 levels are decreased in two HCC samples compared to normal liver tissue [44]. However, GM3 is involved in cell migration, which is an important step in the process of metastasis. Li et al. showed that GM3, but not GM2, inhibited epidermal growth factor-stimulated cell migration and promoted the hepatocyte growth factor-stimulated migration in Hepa1–6 cells (Figure 2A). These effects are mediated through the activation of the PI3K/Akt signaling pathway [46,47,48].

Su et al. showed that ganglioside synthesis is increased in the livers of mice in an animal model featuring activation and expansion of liver progenitor-like cells and liver cancer (stem) cells [49]. Together with elevated ganglioside synthesis, the expansion of mouse hepatic stem/progenitor cells was increased. Furthermore, GM1 ganglioside levels were significantly higher in the *epithelial cellular adhesion molecule* (EpCAM) positive *cancer stem cell* (CSC)-like HCC cell line JHH7 (Table 1 and Figure 2B). *D-threo-1-phenyl-2-decanoylamino-3-morpholino-1-propanol* (D-PDMP), an inhibitor of UGCG, decreased ganglioside synthesis and suppressed cell proliferation and spheroid growth of JHH7cells; whereas apoptotic and necrotic cell death were not impacted [49]. PDMP effects were attributed to decreased expression of Aurora kinase A, Aurora kinase B, protein kinase TKK, kinetochore protein NDC80 homolog, Ki67, and CCNB1; whereas p53 was increased (Figure 2B). Accordingly, p53 may have led to the decrease in expression of the aforementioned cell cycle/cytoskeletal-regulatory proteins. Interestingly, the inhibition of ganglioside synthesis also changed the lipid composition of liver cancer cells [49]. The authors conclude that blocking ganglioside synthesis might be a novel therapeutic option for HCC patients [49]. However, since PDMP blocks UGCG at the first key step in GSL synthesis, the study results from Su et al. do not clearly indicate that the described effect is ascribable to gangliosides or possibly GlcCers.

Interestingly (and contrary to other studies), natrin (purified from snake venom and a member of the *cysteine-rich secretory protein* (CRISP) family) exhibits anticancer activity in HCC by inhibiting cell proliferation and inducing apoptosis. Lu et al. identified gangliosides as potential biomarkers for natrin-induced apoptosis [50]. Following natrin treatment, gangliosides increased in concentration, as well as the ratio of Bax to Bcl-2, which indicates higher susceptibility of cells to apoptosis. Since UGCG/gangliosides are increased in HCC, the question arises whether cells might reach a point of excessive ganglioside accumulation and therefore apoptosis would be induced. Additionally, more research is required about which specific ganglioside species are changed in HCC.

Another important parameter to consider is the immune response in HCC development. Zhu et al. investigated immune responses based on the HCC cohort of The Cancer Genome Atlas (TCGA) database [62]. By assigning immune cell types to three different immunity groups (low, medium, and high) and following *Kyoto Encyclopedia of Genes and Genomes* (KEGG) pathway analysis, the authors showed that the immunity groups differ in gene expression of ganglio series GSL producing proteins [62] (Figure 2B). This poses the question as to whether the immune response (immune cell infiltration of the tumor) can be influenced and therefore immunotherapy enhanced.

Wu et al. showed increased fucosyl GM1 levels in the serum of HCC patients, which indicates a potential prognostic marker function of this ganglioside [40]. However, during induction of rat hepatoma by DEN, GD3 increased in malignant tissues compared to normal tissue, as well as the appearance of precancerous hepatocytes [51]. Furthermore, the ganglioside GD3 induces cell death by targeting mitochondria and engagement of the apoptosome leading to inhibition of survival pathways in HCC, which is reviewed here [52]. However, GD1α is increased in two HCC samples compared to normal liver tissue [44].

CD75s-(Neu5Acα6Galβ4GlcNAcβ3Galβ4Glcβ1Cer) and iso-CD75s-gangliosides (Neu5Acα3Galβ4GlcNAcβ3Galβ4Glcβ1Cer) are increased in HCC tissue, but independent of ST6GAL1 and ST3GAL6 expression [53]. NeuGcGM3 ganglioside is overexpressed in HCC as well [54]; therefore it might be a potential target for HCC therapy. Additionally, ganglioside depletion reduces integrin-mediated cell adhesiveness of rat hepatoma cells [63].

### 4.5. Lacto/Neo-Lacto Series Glycosphingolipids

Zhu et al. showed that fucosylated GSLs are overexpressed in HCC samples compared to the adjacent tissue [41]. In detail, Fuc(Hex)_3_HexNAc-Cer (Fucα_2_Galβ_3_GlcNAcβ_3_Galβ_4_Glcβ_1_Cer, Fucα_2_Galβ_4_GlcNAcβ_3_Galβ_4_Glcβ_1_Cer (H5-2), Fucα_3_(Galβ_4_)GlcNAcβ_3_Galβ_4_Glcβ_1_Cer, Fucα_4_(Galβ_3_)GlcNAcβ_3_Galβ_4_Glcβ_1_Cer), Fuc_2_(Hex)_3_HexNAc-Cer (Fucα_2_Galβ_3_(Fucα_4_)GlcNAcβ_3_Galβ_4_Glcβ_1_Cer (Leb-6), Fucα_2_Galβ_4_(Fucα_3_)GlcNAcβ_3_Galβ_4_Glcβ_1_Cer (Ley-6)), and Fuc(Hex)_4_HexNAc-Cer (Fucα_2_Galβ_3_GalNAcβ_3_Galα_4_Galβ_4_Glcβ_1_Cer) are expressed in HCC samples [41]. More research is needed to shed light on the role of the fucosylated lacto/neo-lacto series GSLs in HCC and in cancer in general.

### 4.6. Sulfatides (Glycosphingolipid Sulfates)

Sulfatides are highly expressed in HCC [55] and regulate integrin αV expression and cell adhesion in hepatoma cells [56]. The regulatory mechanisms are based on complexing of sulfatide with *paired amphipathic helix protein* (SIN3B) leading to reduced binding affinity of SIN3B to *histone deacetylase 2* (HDAC2). Subsequently, HDAC2 recruitment to the integrin αV gene promoter is reduced and the promoter thereby activated. This leads to enhanced tumor metastasis [57] (Table 1).

## 5. Conclusions

UGCG/GlcCer is potentially involved in the development of sorafenib resistance in HCC (Table 1). The complex GSL species globoside, which Globo H belongs to, may serve as a prognostic marker for HCC (Table 1). Gangliosides such as GM1 might act as a potential liver cancer stem cell marker in HCC therapy and sulfatides are involved in tumor metastasis (Table 1). Gaucher disease patients are treated either with an *enzyme replacement therapy* (ERT) or *substrate reduction therapy* (SRT). SRT encompasses treatment with the UGCG inhibitors eliglustat (first-line therapy) [64] or miglustat. Currently, venglustat is under phase III clinical trials for Fabry disease, autosomal dominant polycystic kidney disease, and other diseases [65]. However, the application of UGCG inhibitors in HCC and in general in cancer patients is limited since a systemic UGCG blockage has severe effects on the organism. Inhibition of the key enzyme of the GSL metabolism interrupts the complete GSL production and results in apoptosis. This might be the reason why only a limited number of studies about miglustat and its application in tumors are published.

Only recently Jenneman et al. conducted a study using the UGCG inhibitor Genz-123346 in a colorectal tumor mouse model [66]. Mice fed Genz-123346 developed fewer and smaller tumors and exhibited a lower count of Ki67-positive cells in tumor-free colon crypts. Respectively, Genz-123346 may present a novel therapeutic approach for colorectal cancer. However, it is important to investigate the role of specific GSL species in HCC, which is challenging since the analytical methods, such as LC-MS/MS, are limited in the ability to differentiate between all GLS species. The detection of galactose and glucose fractions in GSLs still needs to be optimized (hexosylceramides). Furthermore, inhibitors for specific GSL species are not available. In the case of the Gb3 synthase (*A4GALT*), the reason is the unknown crystal structure of the protein (reviewed in [67]). A more promising approach for using GSLs in cancer therapy would be to use them for tumor tissue labeling.

Many cancer types overexpress certain GSL species such as GD2 and Gb3. GD2 is almost not expressed in normal tissue (reviewed in [68]). Being shed from cancer cells, GD2 influences immune response. Studies show that GD2 blocking through antibody binding (cancer immunotherapy) leads to apoptotic cell death (reviewed in [68]). Gb3 functions as a receptor for the Shiga toxin. Artificially altered Shiga toxins can be used for labeling the tumor tissue and delivering drugs to the tumor tissues in the patient (reviewed in [69]). Furthermore, antibody-conjugated nanoparticles, targeted against certain GSL species, could also be used for targeted drug delivery. However, the role of GSLs in cancer development, and opportunities to target them or use them as cancer biomarkers remains unclear and warrants further investigation.

## Figures and Tables

**Figure 1 ijms-23-04477-f001:**
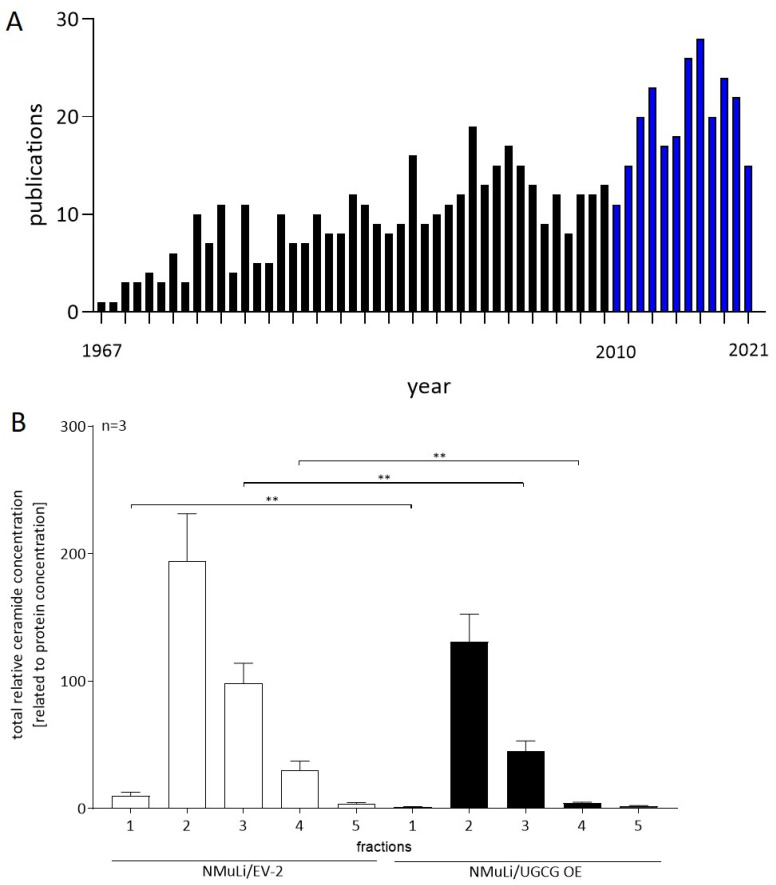
GSLs and HCC related publications and ceramide concentrations in glycosphingolipid-enriched microdomains (GEMs) in NMuLi/UGCG OE and control cells. (**A**) Publications identified by the key word combinations *HCC glucosylceramide*, *HCC glycosphingolipids*, *Gaucher HCC*, *Gaucher liver cancer*, *liver cancer glycosphingolipids*, *hepatocellular carcinoma gangliosides*, *hepatocellular carcinoma globosides*, *hepatocellular carcinoma hexosylceramide*, and *hepatocellular carcinoma* lactosylceramide between 1967 and 2021 [17]. Notably, the majority of these studies were identified by the keywords *liver, cancer,* and *glycosphingolipids*. Blue bars represent years 2010 to 2021 (increase in published studies). (**B**) GEMs were isolated by sucrose density centrifugation and GEMs verified by cholesterol and GlcCer concentration determination. NMuLi/EV-2 = empty vector control; NMuLi/UGCG OE = UGCG overexpressing cells. The GEM verification for these samples is published in [18]. Data are represented as a mean of n = 3 ± SEM. Unpaired t test with Welch’s correction. ** *p* < 0.01.

**Figure 2 ijms-23-04477-f002:**
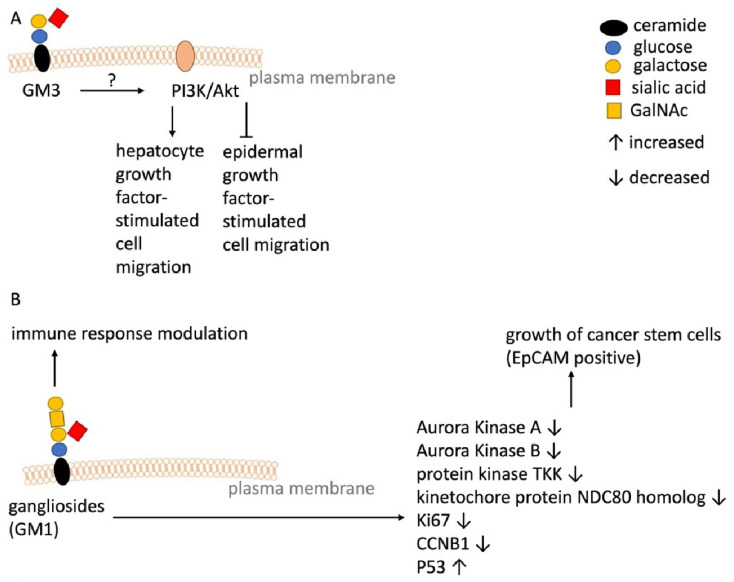
Schematic overview of ganglioside-mediated effects in HCC. (**A**) Ganglioside-mediated effects on cell migration. (**B**) Ganglioside-mediated effects on cancer stem cells and immune response.

**Table 1 ijms-23-04477-t001:** Glycosphingolipid deregulation in HCC.

Glycosphingolipid Species	Change	Effect	References
GlcCer	↑	Unknown	[10,11,35,36]
		tumor development via mTOR	[37]
		Sorafenib resistance	[13]
Lactosylceramide	↑	Unknown	[35,38,39]
DSGG, Gb3, Gb2	↑	Unknown	[40]
Gb3, Gb4	↓	Unknown	[39]
Globo H	↑	Unknown	[41]
		Liver CSC	[42]
Iso-Gb4	↑	Unknown	[43]
GM2	↑	Unknown	[44,45]
GM3	↓	Unknown	[44]
GM3	↑	Cell migration	[46,47,48]
Gangliosides in general	↑	Liver progenitor-like cells and liver CSC	[49]
GM1	↑	In EpCAM positive CSC-like HCC cell line JHH7	[49]
Gangliosides in general	↑	Natrin-induced apoptosis	[50]
Fucosyl GM1	↑	Unknown	[40]
GD3	↑	Inhibition of survival pathways	[51,52]
GD1α	↑	Unknown	[44]
CD75s- and iso-CD75s-gangliosides	↑	Unknown	[53]
NeuGcGM3 ganglioside	↑	Unknown	[54]
Fucosylated GSLs	↑	Unknown	[41]
Sulfatides	↑	Tumor metastasis	[55,56,57]

Table legend: GSLs = glycosphingolipids, CSC = cancer stem cells, EpCAM = epithelial cellular adhesion molecule, HCC = hepatocellular carcinoma, GlcCer = glucosylceramide, mTOR = mechanistic Target of Rapamycin.

## Data Availability

The publications quoted are publicly available from the publishers. Raw data may be requested from the corresponding author.

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
