# Peer review of "Update on Glycosphingolipids Abundance in Hepatocellular Carcinoma"

_ijms, 2022, doi:10.3390/ijms23094477_

Round 1

Reviewer 1 Report

The manuscript "Update on glycosphingolipids abundance in hepatocellular car-2 cinoma" provides a new look at the pathogenesis of lipid disorders.

My comments:

  1. Show the GBA mutations in the Morbus Gaucher.
  2. Present the molecular relationships schematically e.g. for gangliosides.
  3. List the lipids in the table and state their contribution to pathogenesis. The manuscript will be more readable.
  4. Correct editorial and linguistic errors.

Author Response

  1. Show the GBA mutations in the Morbus Gaucher.

We mentioned the GBA mutations and types briefly (line 139-143).

  1. Present the molecular relationships schematically e.g. for gangliosides.

We added Figure 2, which is depicting ganglioside-mediated effects in HCC.

  1. List the lipids in the table and state their contribution to pathogenesis. The manuscript will be more readable.

Table 1 displays glycosphingolipid species, which are deregulated in HCC (line 335).

  1. Correct editorial and linguistic errors.

We corrected all editorial and linguistic errors.

Reviewer 2 Report

This review on glycosphingolipids abundance in hepatocellular carcinoma has a good structure with a logical flow of data and a clear style. The authors provide a good summary of the state of the art.   1. The number of non-standard abbreviations should be drastically reduced. Excessive use of abbreviations is annoying and limits readability. 2. Line 290: typing error in species. In both the original Latin and in English “species” is the spelling of both the singular and plural forms. 3. The activity of UDP-glucose ceramide glycosyltransferase (UGCG) is significantly elevated in several human cancers including breast, cervix, colon, non-small cell lung cancer, and papillary thyroid carcinoma . Its overexpression correlates with chemoresistance. After reading the review, I did not find hard evidence that UGCG is a potential oncotarget. The conclusion of the authors is quite realistic and the final sentence of the paper is the correct representation of the state of the art. It might be opportune to paraphrase this last sentence in the abstract. 4. Reference 67 seems to be incomplete.

Author Response

  1. Show the GBA mutations in the Morbus Gaucher.

We mentioned the GBA mutations and types briefly (line 139-143).

  1. Present the molecular relationships schematically e.g. for gangliosides.

We added Figure 2, which is depicting ganglioside-mediated effects in HCC.

  1. List the lipids in the table and state their contribution to pathogenesis. The manuscript will be more readable.

Table 1 displays glycosphingolipid species, which are deregulated in HCC (line 335).

  1. Correct editorial and linguistic errors.

We corrected all editorial and linguistic errors.

Reviewer 2

  1. The number of non-standard abbreviations should be drastically reduced. Excessive use of abbreviations is annoying and limits readability.

We reduced the number of non-standard abbreviations (ACC, dhCer, GalCer, LacCer, HGF, EGF and ERK) to increase readability. However, glycosphingolipid species are generally abbreviated due to their long names.

  1. Line 290: typing error in species. In both the original Latin and in English “species” is the spelling of both the singular and plural forms.

We corrected the typing error.

  1. The activity of UDP-glucose ceramide glycosyltransferase (UGCG) is significantly elevated in several human cancers including breast, cervix, colon, non-small cell lung cancer, and papillary thyroid carcinoma . Its overexpression correlates with chemoresistance. After reading the review, I did not find hard evidence that UGCG is a potential oncotarget. The conclusion of the authors is quite realistic and the final sentence of the paper is the correct representation of the state of the art. It might be opportune to paraphrase this last sentence in the abstract.

We thank the reviewer for this comment and added the following sentence to the abstract (line 25-26): Overall, the role GSLs play in HCC progression and their ability to serve as biomarkers or prognostic indicators for HCC, requires further investigation.

  1. Reference 67 seems to be incomplete.

We apologies for this and completed the references.